# The expression of MIR125B transcripts and bone phenotypes in *Mir125b2*-deficient mice

Tomohiro Ogasawara[1]*, Shota Ito[1], Shintaro Ogashira[1], Tomonori Hoshino[2], Yusuke Sotomaru[3], Yuji Yoshiko[4], Kotaro Tanimoto[5]

**1** Department of Orthodontics, Division of Oral Health and Development, Hiroshima University Hospital, Hiroshima, Japan, **2** Neuroprotection Research Laboratories, Department of Neurology and Radiology, Massachusetts General Hospital and Harvard Medical School, Charlestown, MA, United States of America, **3** Natural Science Center for Basic Research and Development, Hiroshima University, Hiroshima, Japan, **4** Pi Skovy, Hiroshima, Japan, **5** Department of Orthodontics and Craniofacial Developmental Biology, Hiroshima University Graduate School of Biomedical and Health Sciences, Hiroshima University, Hiroshima, Japan

* t-ogasawara@hiroshima-u.ac.jp

**Data Availability Statement:** All relevant data are within the paper and its Supporting Information files.

## Abstract

MIR125B, particularly its 5p strand, is apparently involved in multiple cellular processes, including osteoblastogenesis and osteoclastogenesis. Given that MIR125B is transcribed from the loci *Mir125b1* and *Mir125b2*, three mature transcripts (MIR125B-5p, MIR125B1-3p, and MIR125B2-3p) are generated (MIR125B-5p is common to both); however, their expression profiles and roles in the bones remain poorly understood. Both primary and mature MIR125B transcripts were differentially expressed in various organs, tissues, and cells, and their expression patterns did not necessarily correlate in wild-type (WT) mice. We generated *Mir125b2* knockout (KO) mice to examine the contribution of *Mir125b2* to MIR125B expression profiles and bone phenotypes. *Mir125b2* KO mice were born and grew normally without any changes in bone parameters. Interestingly, in WT and *Mir125b2* KO, MIR125B-5p was abundant in the calvaria and bone marrow stromal cells. These results indicate that the genetic ablation of *Mir125b2* does not impinge on the bones of mice, attracting greater attention to MIR125B-5p derived from *Mir125b1*. Future studies should investigate *the* conditional deletion of *Mir125b1* and both *Mir125b1* and *Mir125b2* in mice.

## Introduction

MicroRNAs (miRNAs) are small non-coding RNAs that play a central role in silencing gene expression by binding to partially complementary sequences of target mRNAs, causing their translational repression or degradation [1]. Currently, nearly 2,700 mature human miRNAs that may be involved in multiple cellular processes and diseases have been identified (https://www.mirbase.org). One miRNA may regulate many target genes, and likewise, one gene may be targeted by many miRNAs. Moreover, miRNAs play crucial roles in cell-cell communication as extracellular vesicle cargo [2], making them participants in complex processes. Increased levels of MIR125B-5p can be detected in the sera or bones in osteoporotic patients (see for example, [3–6]). Concomitantly, transfection of mouse bone marrow-derived

**Funding:** This study was supported by JSPS KAKENHI Grant Number JP22K17252. Initials of author who received it is S.I. Full name of funding is Japan Society for the Promotion of Science. https://www.jsps.go.jp the funders had no role in study design, data collection and analysis, decision to publish, or preparation of the manuscript.

**Competing interests:** The authors have declared that no competing interests exist.

mesenchymal stem cells (BM-MSCs) with an MIR125B-5p mimic suppressed BMP2-dependent osteogenic differentiation by inhibiting *Cbfb* [7]. Similar results were obtained in human BM-MSCs transfected with an *Mir125b* lentiviral vector targeting *BMPR1B* [8]. An MIR125B-5p mimic also targeted *Traf6* and enhanced the ratio of RANKL/OPG via the JAK2/STAT3 pathway in mouse osteoblastic MC3T3-E1 cells [6]. We previously demonstrated that MIR125B-5p is contained in matrix vesicles budding from MC3T3-E1 cells and that MIR125B-5p targeted *Prdm1*, inhibiting osteoclast formation [9]. Transgenic mice overexpressing *Mir125b1* in osteoblasts exhibited high bone mass with decreased bone resorption while maintaining osteoblastic bone formation [9]. These results suggest that MIR125B-5p is involved in bone metabolism under both physiological and pathological conditions.

Mature MIR125B transcripts are encoded by two different genes in humans: *Mir125b1* located on chromosome (chr)11 (accession #, MI0000446, http://www.mirbase.org/) and *Mir125b2* on chr21 (MI0000470). In mice, the former resides on chr9 (MI0000725), and the latter on chr16 (MI0000152). miRNAs are transcribed into hairpin-containing primary transcripts (pri-miRNAs), which are then processed into precursor miRNAs that are approximately 70 nucleotides long. Eventually, one of the single-stranded miRNAs is generated (termed the "guide strand"), and the other strand (termed the "passenger strand") is generally, but not always, destroyed [10]. The 5p mature strand, MIR125B-5p, is common to *Mir125b1* and *Mir125b2* genes, whereas the 3p strands, MIR125B1-3p and MIR125B2-3p, are different from each other. However, the functional roles of *Mir125b* genes remain largely unknown. Thus. We generated *Mir125b1* and *Mir125b2* knockout (KO) mice. Although homozygotes of the former were embryonically lethal, those of the latter were born normally. We examined the expression profiles of MIR125B transcripts in wild-type (WT) mice and the impact of *Mir125b2* deficiency on their expression and bone phenotype.

## Materials and methods

### Mice

*Mir125b2* KO mice were generated with a C57BL/6J background using the CRISPR/Cas9 System. Two single guide RNAs (`ACTCTAATTCCCAAGCTGTC` and `AACAGGCATAGATTCTGCAT`, 32.15 ng/μL each; Thermo Fisher Scientific, Waltham, MA, USA) were co-injected with Cas9 mRNA (62.5 ng/μL; SBI, Palo Alto, CA, USA) into zygotes, according to the manufacturer's instructions, to target the *Mir125b2* coding region. In the mouse *Mir125b2* gene (NCBI Gene database), the target genomic locus did not contain any known coding sequences. The founder lines were genotyped using PCR (see the primer set in S1 Table) and DNA sequence analysis. Among the several lines of *Mir125b2* KO mice (homozygotes) obtained, we chose *Mir125b2* KO line #2, which carried a 199 bp deletion (S1 Fig) based on their similar appearance, live birth, and body weight changes. Animal use and procedures were approved by the Institutional Animal Care and Use Committee of the Central Institute for Experimental Animals and the Committee of Animal Experimentation at Hiroshima University (#A20-3-4).

### Serum biochemical analysis

Sera were obtained from 12- to 14-week-old male WT and *Mir125b2* KO mice and analyzed using a biochemical automatic analyzer (7180 Clinical Analyzer; Hitachi High-Tech, Tokyo, Japan).

### RNA isolation and real-time RT-PCR

Total RNA was extracted from crushed frozen tissues of 12-week-old male mice and cultured using RNAiso Plus (Takara Bio, Shiga, Japan). The miReasy Mini Kit (QIAGEN, Hilden,

Germany) was used according to the manufacturer's instructions. Serum small RNAs were isolated using the miRNeasy Serum/Plasma Advanced Kit (QIAGEN). Genomic DNA was digested using Dnase I (QIAGEN), and cDNA was synthesized using M-MuLV Reverse Transcriptase (New England BioLabs, Ipswich, MA) and ReverTra Ace (TOYOBO, Osaka, Japan) for miRNAs and mRNAs/pri-miRNAs, respectively. Quantitative real-time PCR was performed using FastStart Universal Probe Master (Roche, Basel, Switzerland) with the TaqMan MicroRNA Assay (Thermo Fisher Scientific) for mature miRNAs, according to the manufacturer's recommendations. THUNDERBIRD Next SYBR qPCR Mix (TOYOBO) was used with primer sets for pri-miRNAs and *Actb* (S1 Table). The TaqMan Pri-miRNA Assay was used for the quantitative analysis of pri-miRNAs (Thermo Fisher Scientific). *Rnu6* and *Actb* were used as internal controls to normalize the miRNA and pri-miRNA levels, respectively. A *Caenorhabditis elegans* MIR39 mimic (Thermo Fisher Scientific) was used to normalize serum sRNA levels [11]. All analyses were performed using the comparative Ct method with a StepOnePlus real-time PCR system (Thermo Fisher Scientific).

## Micro-computed tomography (μCT) analysis

Left tibiae were dissected from 12-week-old male mice, fixed in 4% paraformaldehyde (PFA) in PBS (FUJIFILM Wako Pure Chemical, Osaka, Japan) at 4°C for 24 h and stored in 70% ethanol at 4°C until use. Whole tibiae were scanned using SkySkan 1176 (Bruker, Kontich, Belgium), and the datasets were reconstructed using Nrecon (v1.7.4.6, Bruker), followed by alignment using Data Viewer (v1.5.6.2, Bruker). The scanning and reconstruction conditions are listed in S2 Table. CTAn (v.1.20.8.0, Bruker) was used to measure bone parameters in the trabecular bone (1.0 mm in width, 0.5 mm below the growth plate). The tibial length was measured from the proximal end of the superior articular surface to the distal end of the medial malleolus.

## Bone histomorphometry

Calcein (10 mg/kg body weight) was injected intraperitoneally twice to 12-week-old male mice at an interval of 5 d. Tibiae were collected 2 d after the second calcein administration and fixed in 4% PFA in PBS for 24 h. After washing with PBS, the fixation solution was replaced with 70% ethanol. The tibiae were then stained with Villanueva bone stain, dehydrated using a graded series of ethanol, and embedded in methyl methacrylate. Plastic sections (5 μm thickness) were prepared, and histometric analysis was performed using a semiautomatic graphic system (Histometry RT CAMERA, System Supply, Nagano, Japan).

## Bone marrow stroma cell (BMSC) and bone marrow macrophage (BMM) cultures

Bone marrow cells (BMCs) were isolated from the femurs and tibiae of 10–15-week-old male mice as described previously [12]. Briefly, after removing soft tissues, both epiphysial ends were cut, and BMCs were flushed out using αMEM (Thermo Fisher Scientific) containing antibiotics (100 U/ml penicillin and 100 μg/ml streptomycin (Sigma Aldrich, St. Louis, MO, USA)). Cells from 2 to 5 mice were pooled, suspended, and passed through a cell strainer (70 μm; Corning, Glendale, AZ, USA). Resuspended cells were cultured in 10 cm dishes with αMEM supplemented with 20% FBS (Sigma Aldrich) and antibiotics (growth medium) at 37°C in humidified 5% $CO_2$ with hypoxia (5% oxygen) [13]. The next day, nonadherent cells were carefully removed by washing, and the cells were maintained for an additional 10 d to obtain osteogenic BMSCs.

To obtain the osteoclast progenitor BMMs, BMCs were hemolyzed in ACK lysis buffer (0.15 M NH$_4$Cl, 0.01 M KHCO$_3$, and 1 mM Na$_2$EDTA, pH 7.4), as previously described [9]. After gentle agitation for 2 min, cells were rinsed and cultured in nonadherent cell culture dishes with αMEM supplemented with 10% FBS and 200 ng/mL M-CSF (PeproTech, Cranbury, NJ) for 2 d at 37°C in humidified 5% CO$_2$. After removing nonadherent cells by washing, BMMs were collected.

### Statistical analysis

Data are expressed as mean ± standard deviation. Statistical analyses were performed using GraphPad Prism 9.2.0 (GraphPad Software, San Diego, CA, USA). The Mann–Whitney U test was used to compare the means between the two groups. Differences between more than two groups were examined using the Kruskal–Wallis test, followed by Dunn's multiple comparison test. Statistical significance was set at $P<0.05$.

## Results

### The expression profiles of MIR125B transcripts in WT mice

We first evaluated levels of *pri-Mir125b1* and *pri-Mir125b2* in the brain, heart, liver, lung, kidney, spleen, testis, skeletal muscle, inguinal adipose tissue, femur (including bone marrow), and calvaria (Fig 1A). Both pri-miRNAs were most abundant in the brain and heart, followed

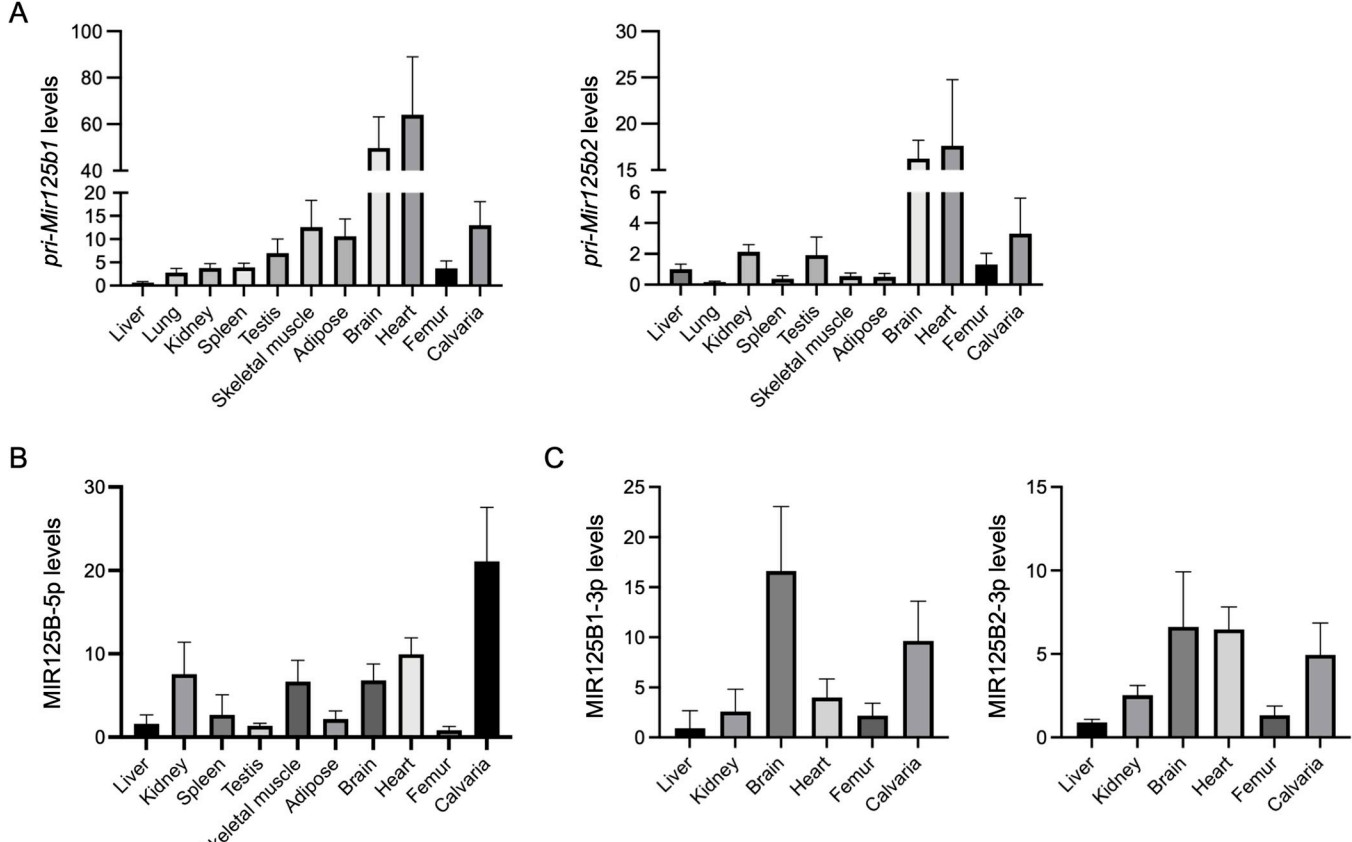

**Fig 1. Transcriptional profiling of *Mir125b* genes in wild-type (WT) mice.** Relative levels of *pri-Mir125b1* and *pri-Mir125b2* (A), MIR125B-5p (B), MIR125B1-3p, and MIR125B2-3p (C) in various organs/tissues of 12-week-old male mice are shown. *Actb* (A) and *Rnu6* (B, C) were used as internal controls. *n* = 4–6.

by calvaria, skeletal muscle, and adipose tissue for *pri-Mir125b1* and the calvaria, kidney, and testis for *pri-Mir125b2*. In bones, *pri-Mir125b1* and *pri-Mir125b2* were more abundant in the calvaria than in the femur, including in BMCs. We then examined MIR125B-5p expression in most of the above organs/tissues and found that MIR125B-5p levels were highest in the calvaria (Fig 1B). MIR125B1-3p and MIR125b2-3p, previously identified in the rat heart [14], human brain [15], kidney [16], and liver [17], were also found to be among the most abundant in organs/tissues (Fig 1C). These results suggest that *Mir125b1* and *Mir125b2* may contribute to the levels of three mature MIR125B transcripts in an organ/tissue-dependent manner and that all mature MIR125B transcripts may be abundant in the calvaria.

## General features of *Mir125b2* KO mice

We used CRISPR/Cas9-mediated genome editing to delete *Mir125b2* and designed guide RNAs. The genomic sequences of the mutant F0 mice are shown in S1 Fig. These mice were further mated with C57BL/6J mice, thereby obtaining F1 mutant mice, before their genotypes were confirmed using PCR product sequencing. *Mir125b2* KO mice (line #2, homozygotes) were born normally at the expected Mendelian ratio (WT, 16.73±11.20; KO, 14.34±10.41 in the interbreeding of heterozygous mice). The appearance of *Mir125b2* KO mice was identical to that of WT mice (Fig 2A). There were no significant differences in body weight changes between WT and *Mir125b2* KO mice in both males and females (Fig 2B). Serological findings in WT or *Mir125b-2* KO male mice are summarized in Table 1.

## The expression profiles of MIR125B transcripts in *Mir125b2* KO mice

We demonstrated the effect of *Mir125b2* deficiency on levels of MIR125B transcripts in representative organs and tissues, including the bones. Concomitant with *Mir125b2* deficiency, *pri-Mir125b2* (S2A Fig) and MIR125B2-3p (S2B Fig) were undetectable in *Mir125b2* KO mice. This reflects the lower levels of MIR125B-5p in the heart, kidneys, spleen, liver, testes, and femurs of *Mir125b2* KO than in WT mice (Fig 3A). Interestingly, despite *Mir125b2* deficiency, MIR125B-5p levels in the brain, calvaria, and serum of *Mir125b2* KO mice were comparable to those of WT mice (Fig 3A). We found no significant differences in MIR125B1-3p levels in the heart, brain, or bones of *Mir125b2* KO and WT mice (Fig 3B), whereas *pri-Mir125b1* levels were unexpectedly suppressed in *Mir125b2* KO hearts and femurs (Fig 3C).

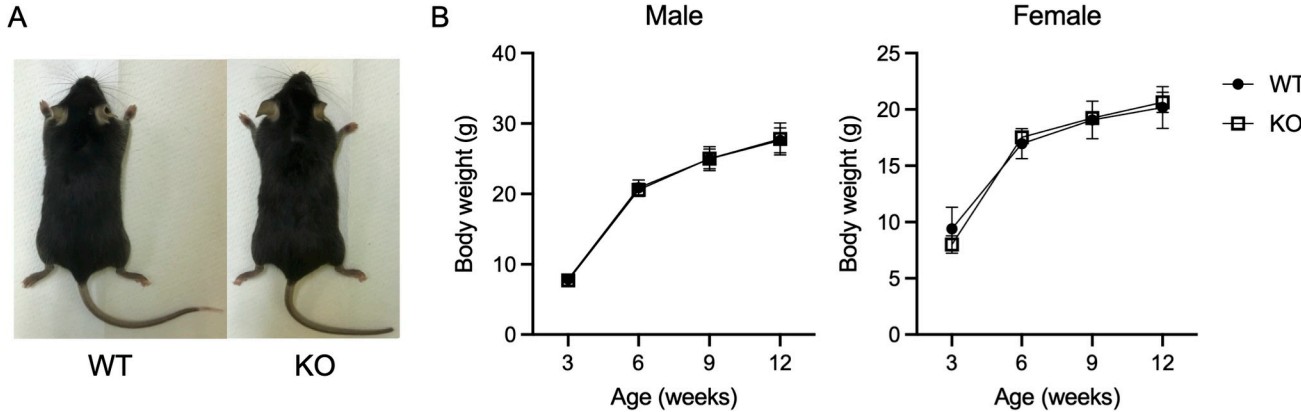

**Fig 2. The gross appearance of WT and *Mir125b2* KO mice and their growth curves.** (A) Representative images of 12-week-old male mice. (B) Body weight changes in mice from 3–12 weeks of age. *n* = 15−18 (males) and 7−9 (females).

**Table 1.  Serum biochemical analysis of WT and *Mir125b2* KO male mice.**

| Parameters | WT | | *Mir125b2* KO | | P value |
|---|---|---|---|---|---|
| | Mean | SD | Mean | SD | |
| TP (g/dL) | 5.200 | 0.1826 | 5.114 | 0.1773 | 0.5402 |
| ALB (g/dL) | 3.243 | 0.1272 | 3.243 | 0.1813 | 0.6096 |
| BUN (mg/dL) | 23.70 | 2.499 | 21.93 | 3.931 | 0.3357 |
| CRE (mg/dL) | 0.1171 | 0.009512 | 0.1100 | 0.01265 | 0.3275 |
| Na (mEq/L) | 154.6 | 2.507 | 156.4 | 1.718 | 0.1521 |
| K (mEq/L) | 4.314 | 0.3288 | 4.286 | 0.5113 | 0.9732 |
| Cl (mEq/L) | 105.4 | 1.272 | 103.9 | 5.014 | 0.3998 |
| Ca (mg/dL) | 9.143 | 0.1902 | 9.357 | 0.4721 | 0.2896 |
| IP (mg/dL) | 10.46 | 1.938 | 10.87 | 1.525 | 0.8788 |
| AST (IU/L) | 103.0 | 40.11 | 141.4 | 73.61 | 0.4557 |
| ALT (IU/L) | 29.57 | 9.289 | 30.86 | 9.263 | 0.5565 |
| LDH (IU/L) | 582.9 | 104.6 | 549.0 | 249.0 | 0.7104 |
| AMY (IU/L) | 2080 | 319.6 | 2247 | 544.7 | 0.9015 |
| T-CHO (mg/dL) | 70.86 | 14.40 | 78.86 | 11.78 | 0.2593 |
| TG (mg/dL) | 23.86 | 13.91 | 24.57 | 14.34 | 0.7389 |
| HDL-C (mg/dL) | 41.29 | 10.29 | 45.00 | 8.622 | 0.3293 |
| T-BIL (mg/dL) | 0.1286 | 0.06466 | 0.1086 | 0.04634 | 0.5554 |
| GLU (mg/dL) | 160.3 | 64.32 | 175.9 | 48.31 | 0.3176 |

WT, wild-type; KO, knockout; SD, standard deviation; TP, total protein; ALB, albumin; BUN, blood urea nitrogen; CRE, creatinine; Cl, chloride; Na, sodium; K, potassium; Ca, calcium; IP, inorganic phosphate; AST, aspartate aminotransferase; ALT, alanine aminotransferase; LDH, lactate dehydrogenase; AMY, amylase; T-CHO, total cholesterol; TG, triglycerides; HDL-C, high-density lipoprotein cholesterol; T-BIL, total bilirubin; GLU, glucose

## Bone morphometric parameters in *Mir125b2* KO mice

To confirm the effect of *Mir125b2* deletion on bone, μCT scans of tibiae were obtained from WT and *Mir125b2* KO mice. No statistical differences between the two genotypes were found in long axis length, bone mineral density, bone volume in the proximal metaphysis, or other trabecular bone parameters (Fig 4A). We further demonstrated the histomorphometry of trabecular bones in WT versus *Mir125b2* KO tibiae. Microscopic images with Villanueva staining did not exhibit any remarkable differences in the growth plate, trabecular bone, or marrow cells between the two genotypes (Fig 4B). Quantitative histomorphometry also revealed that parameters such as the number of osteoblasts and osteoclasts, mineral apposition rate, and eroded surface/bone surface in *Mir125b2* KO mice were not significantly different from those in WT mice (Fig 4C and S3 Fig).

## BMSCs and BMMs in *Mir125b2* KO mice

To further clarify the effects of *Mir125b2* deficiency on bone, BMCs and their corresponding BMSCs were isolated from *Mir125b2* KO and WT mice. Regardless of *Mir125b2* deficiency, MIR125B-5p was equally present in BMCs and BMSCs in WT and *Mir125b2* KO mice, and its levels were significantly higher in BMSCs than in BMCs (Fig 5). These expression patterns were the same in MIR125B1-3p. MIR25B2-3p was undetectable in *Mir125b2* KO BMCs and BMSCs, whereas it was relatively abundant in WT BMSCs (Fig 5). Levels of MIR125B-5p and MIR125B1-3p in BMMs were lower or undetectable in both genotypes (Fig 5B).

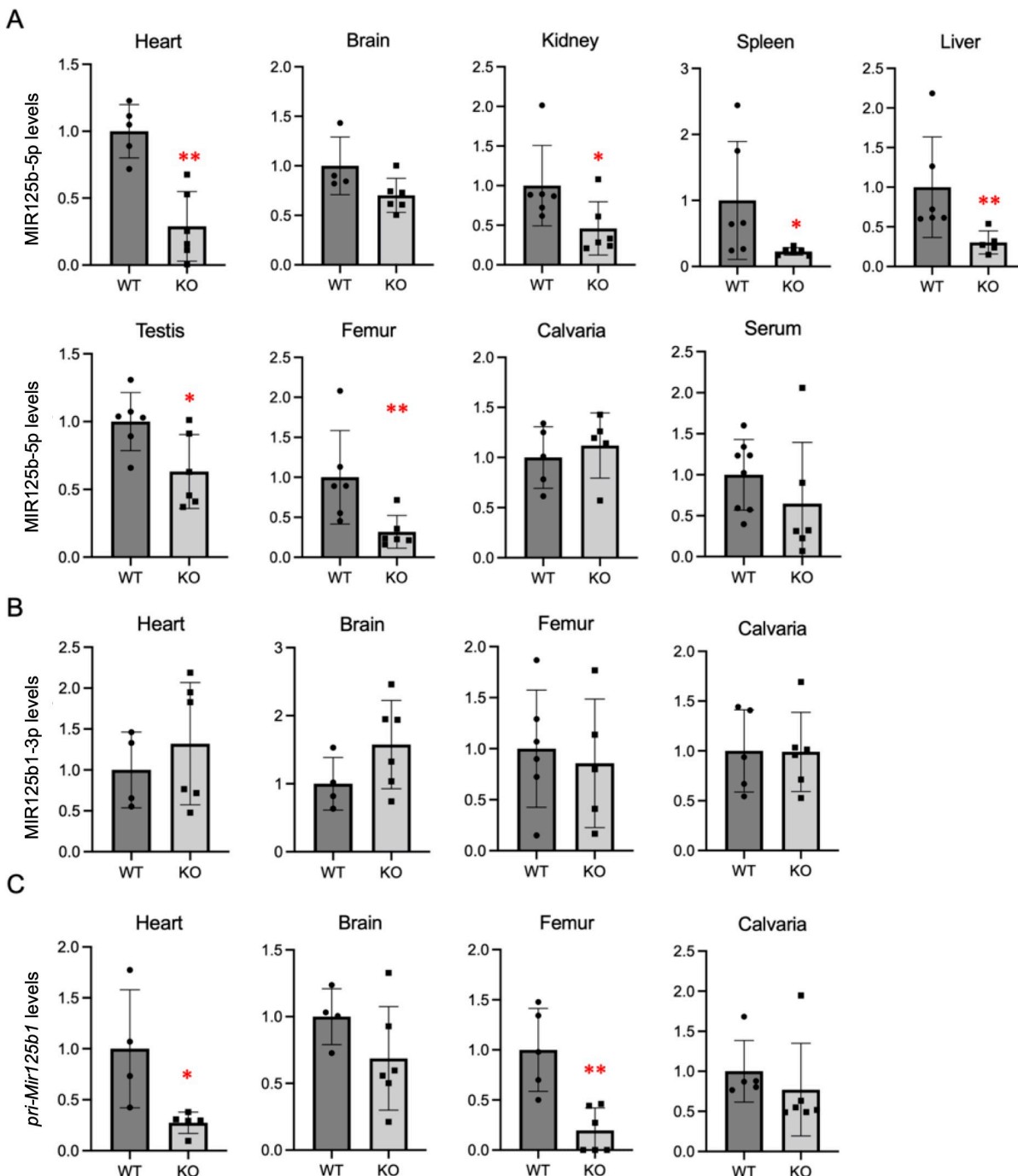

**Fig 3. Levels of MIR125B-5p and MIR125B1 transcripts in WT and *Mir125b2* KO mice.** Relative levels of MIR125B-5p (A), MIR125B1-3p (B), and *pri-Mir125b1* (C) in various organs/tissues of 12-week-old male mice are shown. Mean values of the WT group were set at 1.0. *Rnu6* (A, B) and *Actb* (C) were used as internal controls. *n* = 4–6. *, *p* < 0.05 and **, *p* < 0.01 versus WT mice.

## Discussion

To date, MIR125B-5p has been detected in the lungs [18, 19], brain [20, 21], heart [22, 23], liver [24], testis [25], adipose tissue [26], bone [9], and skeletal muscles [27] of mice. However, little information is available regarding the expression profiles of MIR125B transcripts and

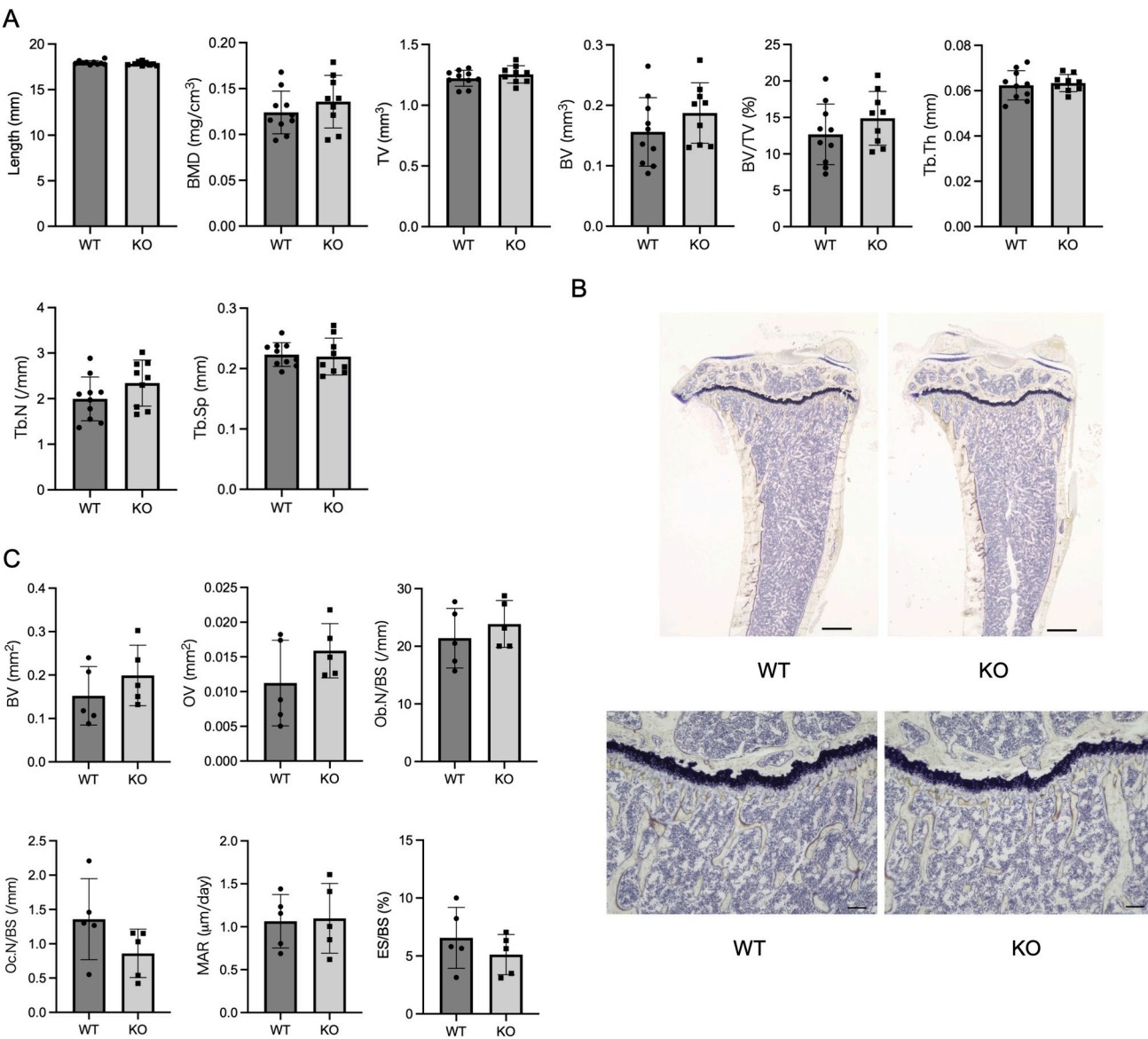

**Fig 4. μCT and histomorphometric analyses of WT and *Mir125b2* KO mouse tibiae.** (A) Trabecular bone parameters in 12-week-old male mice were assessed using μCT. BMD, bone mineral density; TV, tissue volume; BV, bone volume; BV/TV, bone volume/tissue volume; Tb.N, trabecular number; Tb. Th, trabecular thickness; Tb.Sp, trabecular separation. *n* = 9–10. (B) Representative images of longitudinal sections with Villanueva staining are shown. The scales in the upper and lower panels represent 500 μm and 100 μm, respectively. (C) Trabecular bone parameters were assessed by bone histomorphometry. BV, bone volume; OV, osteoid volume; Ob. N/BS, number of osteoblasts per bone surface; Oc. N/BS, number of osteoclasts per bone surface; MAR, mineral apposition rate; ES/BS, eroded surface per bone surface. *n* = 5.

their genetic origins. Mouse MIR125B transcripts in the TissueAtlas database (https://ccb-web.cs.uni-saarland.de/tissueatlas2) were only available for a limited number of organs and tissues. To our knowledge, this is the first study to demonstrate the expression patterns of three mature MIR125B transcripts and their pri-miRNAs in the major mouse organs or tissues. There have also been no reports on our finding that bone cells, especially BMSCs, abundantly express mature MIR125B transcripts. Our previous study demonstrated that among the MIR125B transcripts, MIR125B-5p was identified by miRNA microarray analysis of matrix vesicles obtained from mouse osteoblastic MC3T3-E1 cells [9], although MIR125B1-3p and

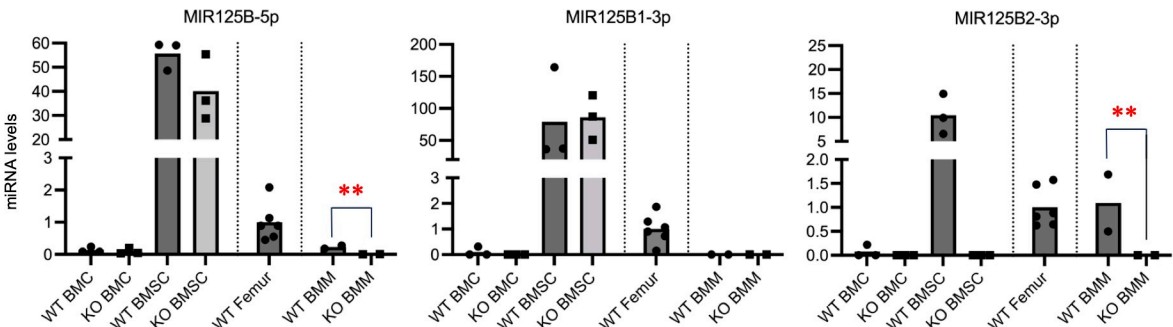

**Fig 5. Levels of mature MIR125B transcripts in BMCs, BMSCs, and BMMs of WT and *Mir125b2* KO mice.** Bone marrow stroma cells (BMSCs) and bone marrow macrophage (BMMs) were obtained from bone marrow cells (BMCs) of 10–15-week-old male mouse femurs and tibiae. Relative levels of MIR125B-5p, MIR125B1-3p, and MIR125B2-3p in BMCs, BMSCs, and BMMs are shown. Femurs of 12-week-old WT male mice were used for comparison. *Rnu6* was used as internal control.

-2-3p were highly expressed in calvaria and BMSCs in this study. As in chondrocyte matrix vesicles [28], this suggests a yet-undefined mechanism underlying the selective enrichment of mature MIR125B transcripts in matrix vesicles.

Li *et al.* found a reduction in litter size in *Mir125b2* KO mice, with a reduced number of sperm cells [21]; however, we could not obtain results supporting this. As described for *Mir125b2* KO mice lacking GACCCTA from the region of MIR125B2 transcripts [22], we did not observe obvious alterations in the appearance and biochemical parameters of male or female *Mir125b2 KO* mice. The *Mir125b2* locus overlaps only with *Mir99ahg*, a long non-coding RNA, not only on mouse chr16 but also on human chr21 (NCBI Gene database). Although a few studies described MIR99AHG as a tumor suppressor and promoter [29–31], the roles of MIR99AHG have remained largely unknown.

The expression patterns of all three mature MIR125B transcripts and their pri-miRNAs in multiple organs, tissues, and cells did not correlate exactly with each other, suggesting that the transcriptional and post-transcriptional regulation of *Mir125b1* and *Mir125b2* genes may depend on the organ, tissue, or cell in question. Decreased MIR125B-5p levels in the heart, kidneys, spleen, liver, testes, and femurs of *Mir125b2* KO mice indicate that cells in these organs predominantly express *pri-Mir125b2* rather than *pri-Mir125b1*. In contrast, *Mir125b1* rather than *Mir125b2* may contribute to MIR125B-5p levels in BMSCs, calvaria, and blood. These results may be linked to the lack of differences in tibial trabecular bone parameters between *Mir125b2* KO and WT mice. *Mir125b2 KO* mice neither mirrored the high bone mass seen in Tg mice [9] nor replicated changes in the osteoblastogenesis of human and mouse osteogenic cells transfected with MIR125B or its inhibitor [6–8, 32]. The lower levels of *pri-Mir125b1* in the heart and femur of *Mir125b2* KO vs. WT mice remain unclear. A similar tendency was reported for *pri-MiR125b2* levels in cardiac-specific *Mir125b1* KO mice [19], implying the necessity for future investigation of the link between *Mir125b1* and *Mir125b2*.

*Ex vivo* studies on BMSC and BMM cultures may help to better understand the exact roles of mature MIR125B transcripts in bone. Levels of MIR125B-5p and MIR125B1-3p not only in WT BMSCs but also in *Mir125b2* KO BMSCs were remarkably higher than those in WT femurs, BMCs, and BMMs. MIR125B-5p may be the most abundant of the three mature MIR125B transcripts in BMSCs, assuming that their PCR amplification efficiencies were nearly equal (MIR125b-5p in WT, Ct = 22–26 versus MIR125B-1-3p and -2-3p in WT, Ct = 34–40). These results suggest that *Mir125b2* and MIR125B2-3p are not involved in osteoblastogenesis. We found that BMMs showed barely detectable levels of the three mature MIR125B transcripts with or without *Mir125b2* deficiency (see also our previous data on

MIR125B-5p, [9]), suggesting that MIR125b expressed in BMMs may not participate in osteoclastogenesis. Of the three mature MIR125B transcripts, MIR125B-5p has drawn attention for its relevance in several diseases, including cardioprotection after acute myocardial infarction [33], progression of renal cell carcinoma [34], resistance of hepatocellular carcinoma to trans-arterial chemoembolization [35], and promotion of osteoporosis [3, 4]. Serum levels of MIR125B-5p have also been implicated in some diseases of the liver [36–38], heart/kidney [39, 40], and bone [3–5]. Taken together with our data and previous studies showing that exogenous application of MIR125B-5p suppressed the osteogenic differentiation of human BMSCs [8, 32] and C3H/10T1/2 cells, a mouse mesenchymal cell line [7], these findings suggest that MIR125B-5p, rather than MIR125B2-3p, impacts bone formation.

The fact that the phenotypic analysis of bone structure in vivo was performed only in the tibia may have made the evaluation of bone structure in *Mir125b2* KO mice localized and is one of the limitations of this study. To gain further insight into *Mir125b2* KO mice with organ- or tissue-specific expression patterns, systemic bone structure needs to be evaluated (e.g., vertebrae, skull, and femur). Furthermore, future studies should include a research agenda for conditional deletion of *Mir125b1* and both *Mir125b1* and *Mir125b2* in mice.

## Supporting information

**S1 Fig. The alignment of genomic DNAs of *Mir125b2* KO mouse lines.** The top line indicates a wild-type mouse sequence (UCSC Genome Browser, https://genome.ucsc.edu/) as a reference, and the others are sequences from F0 mice in Line 1–Line 9. The sequences of the *Mir125b2* precursor, PAM, and gRNA are enclosed within the frame.
(TIFF)

**S2 Fig. Levels of *pri-Mir125b2* and MIR125B2-3p in WT and *Mir125b2* KO mice.** Relative levels of *pri-Mir125b2* (A) and MIR125B2-3p (B) in the heart, brain, femurs, and calvaria of 12-week-old male mice. The mean values of the WT groups were set to 1.0. *Actb* (A) and *Rnu6* (B) were used as internal controls. $n = 4–6$. *, $p < 0.05$ and **, $p < 0.01$ versus WT mice.
(TIFF)

**S3 Fig. Histomorphometric parameters of WT and *Mir125b2* KO mouse tibiae.** See also Fig 4(C). TV, tissue volume; BS, bone surface; OS, osteoid surface; Ob.S, osteoblast surface; Oc.S, osteoclast surface; Ob.S/BS, osteoblast surface/bone surface; Oc.S/BS, osteoclast surface/bone surface; Tb.Th, trabecular thickness; Tb.N, trabecular number; Tb.Sp, trabecular separation; ES, eroded surface; MS/OS, mineralized surface/osteoid surface; LS/OS, labeled surface/osteoid surface; BFR/BS, bone formation rate/bone surface; BFR/TV, bone formation rate/tissue volume. $n = 5$.
(TIFF)

**S1 Table. Primer sets for PCR.**
(DOCX)

**S2 Table. μCT scanning and reconstruction conditions.**
(DOCX)

## Author Contributions

**Conceptualization:** Yuji Yoshiko.

**Data curation:** Tomohiro Ogasawara.

**Formal analysis:** Tomohiro Ogasawara.

**Funding acquisition:** Shota Ito.

**Investigation:** Tomohiro Ogasawara, Shota Ito, Shintaro Ogashira, Yusuke Sotomaru.

**Methodology:** Yuji Yoshiko.

**Project administration:** Yuji Yoshiko.

**Resources:** Yusuke Sotomaru, Yuji Yoshiko.

**Supervision:** Kotaro Tanimoto.

**Validation:** Yuji Yoshiko.

**Visualization:** Tomohiro Ogasawara.

**Writing – original draft:** Tomohiro Ogasawara.

**Writing – review & editing:** Tomohiro Ogasawara, Shota Ito, Shintaro Ogashira, Tomonori Hoshino, Yusuke Sotomaru, Yuji Yoshiko, Kotaro Tanimoto.

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
