## [Decision Letter · Decision Letter 0]

22 Mar 2024

PONE-D-24-05753The expression of MIR125B transcripts and bone phenotypes in Mir125b2 deficient micePLOS ONE

Dear Dr. Ogasawara,

Thank you for submitting your manuscript to PLOS ONE. After careful consideration, we feel that it has merit but does not fully meet PLOS ONE’s publication criteria as it currently stands. Therefore, we invite you to submit a revised version of the manuscript that addresses the points raised during the review process.

We look forward to receiving your revised manuscript.

Kind regards,

Gary S. Stein

Academic Editor

PLOS ONE

Journal Requirements:

   "This study was supported by JSPS KAKENHI Grant Number JP22K17252. Initials of author who received it is S.I. Full name of funding is Japan Society for the Promotion of Science. 

https://www.jsps.go.jp"

Reviewers' comments:

Reviewer's Responses to Questions

**Comments to the Author**

1. Is the manuscript technically sound, and do the data support the conclusions?

Reviewer #1: Yes

2. Has the statistical analysis been performed appropriately and rigorously? 

Reviewer #1: Yes

3. Have the authors made all data underlying the findings in their manuscript fully available?

Reviewer #1: Yes

4. Is the manuscript presented in an intelligible fashion and written in standard English?

Reviewer #1: Yes

5. Review Comments to the Author

Reviewer #1: This paper investigates the role of Mir125b2 in the bone phenotype of 3 months old mice. Using CRISPR/Ca9system the authors delete Mir125b2 globally and convincingly show the deletion with mRNA. . KO male and female mice have the same weight than respective WT control and grow normally. Deletion of Mir125b2 does not cause any bone phenotype as indicated by the analysis of the tibua by micro-CT and histomorphometry in male mice The analysis is detailed and presented appropriately.

Minor issues:

The paper would benefit y the review of an English professional service.

The absence of the bone phenotype should be confirmed in another bone such as vertebra or femur.

Overall, this paper clearly conveys the message and does not have any major issues.

6. PLOS authors have the option to publish the peer review history of their article (what does this mean?). If published, this will include your full peer review and any attached files.

Reviewer #1: No

---

## [Author Response · Author response to Decision Letter 0]

2 May 2024

Response to Reviewers

［Response］： Thank you for your insightful suggestion. Accordingly, the manuscript has been rechecked and the necessary changes have been made throughout the manuscript. 

 "This study was supported by JSPS KAKENHI Grant Number JP22K17252. Initials of author who received it is S.I. Full name of funding is Japan Society for the Promotion of Science. 

https://www.jsps.go.jp"

Please state what role the funders took in the study.

［Response］： Thank you for your insightful advice. The funders had no role in the design of the study, so we have stated the following in the manuscript: "The funders had no role in study design, data collection and analysis, decision to publish, or preparation of the manuscript." 

［Response］： Thank you for your insightful suggestion. According to your suggestion, the reference list was carefully rechecked and slight formatting corrections have been made.

4. Reviewer #1: This paper investigates the role of Mir125b2 in the bone phenotype of 3 months old mice. Using CRISPR/Ca9system the authors delete Mir125b2 globally and convincingly show the deletion with mRNA. . KO male and female mice have the same weight than respective WT control and grow normally. Deletion of Mir125b2 does not cause any bone phenotype as indicated by the analysis of the tibua by micro-CT and histomorphometry in male mice The analysis is detailed and presented appropriately.

Minor issues:

The paper would benefit the review of an English professional service.

［Response］： Thank you for your insightful suggestion. We have revised the manuscript. In addition, the manuscript has been proofread by a native English speaker specialized in the editing of scientific manuscripts. 

The absence of the bone phenotype should be confirmed in another bone such as vertebra or femur.

［Response］： We are grateful for your suggestions. We collected both the femur and tibia and took μCT. The ROI had to be strictly defined so as not to miss microscopic changes, we determined by setting the long axis of the bone and using the distance from the growth plate.The femoral growth plate is not flat, and we were concerned about errors in distance measurement, so we employed CT data from the tibia. Due to methodological limitations, CT data from the femur and vertebrae were not employed, but the possibility that we could only assess the phenotype of KO mice locally is noted as a limitation in the manuscript.

---

## [Editor Report · Decision Letter 1]

7 May 2024

The expression of MIR125B transcripts and bone phenotypes in Mir125b2-deficient mice

PONE-D-24-05753R1

Dear Dr. Ogasawara,

We’re pleased to inform you that your manuscript has been judged scientifically suitable for publication and will be formally accepted for publication once it meets all outstanding technical requirements.

Kind regards,

Gary S. Stein

Academic Editor

PLOS ONE

---

## [Editor Report · Acceptance letter]

13 Jun 2024

PONE-D-24-05753R1 

PLOS ONE

Dear Dr. Ogasawara, 

I'm pleased to inform you that your manuscript has been deemed suitable for publication in PLOS ONE. Congratulations! Your manuscript is now being handed over to our production team.

Kind regards, 

on behalf of

Dr. Gary S. Stein 

Academic Editor

PLOS ONE